# Observation of spin-glass-like characteristics in ferrimagnetic TbCo through energy-level-selective approach

Ji-Ho Park[1,5], Won Tae Kim[1,5], Woonjae Won[1], Jun-Ho Kang[1], Soogil Lee [2], Byong-Guk Park [2], Byoung S. Ham [3], Younghun Jo[4], Fabian Rotermund [1] ✉ & Kab-Jin Kim [1] ✉

Rare earth (RE)–transition metal (TM) ferrimagnetic alloys are gaining increasing attention because of their potential use in the field of anti-ferromagnetic spintronics. The moment from RE sub-lattice primarily originates from the $4f$-electrons located far below the Fermi level ($E_F$), and the moment from TM sub-lattice arises from the $3d$-electrons across the $E_F$. Therefore, the individual magnetic moment configurations at different energy levels must be explored to clarify the microscopic mechanism of anti-ferromagnetic spin dynamics. Considering these issues, here we investigate the energy-level-selective magnetic moment configuration in ferrimagnetic TbCo alloy. We reveal that magnetic moments at deeper energy levels are more easily altered by the external magnetic field than those near the $E_F$. More importantly, we find that the magnetic moments at deeper energy levels exhibit a spin-glass-like characteristics such as slow dynamics and magnetic moment freezing whereas those at $E_F$ do not. These unique energy-level-dependent characteristics of RE-TM ferrimagnet may provide a better understanding of ferrimagnet, which could be useful in spintronic applications as well as in spin-glass studies.

Ferrimagnets have attracted renewed interest owing to the recent advances in antiferromagnetic spintronics[1-5] as well as their unique advantages. In particular, they microscopically exhibit a spin config-uration with antiferromagnetic exchange coupling but have an overall finite net magnetic moment. As a result, the antiferromagnetically coupled spin configuration can be controlled by an external magnetic field, which has not been achieved in pure antiferromagnets yet. Spe-cifically, rare earth (RE)–transition metal (TM) ferrimagnetic alloys provide an opportunity to observe pure antiferromagnetic spin dynamics because RE–TM ferrimagnets have a magnetization com-pensation temperature $T_M$ and an angular momentum compensation temperature $T_A$ owing to the different Landé g-factors between the RE

and the TM[6-8]. Indeed, pure antiferromagnetic spin dynamics have been demonstrated at $T_A$, where the net angular momentum vanishes; however, the net magnetic moment is finite[9-11]. This interesting feature of RE–TM ferrimagnets has thus far been explained by considering two moments from RE and TM sub-lattices on an equal footing[12-17]. How-ever, this assumption is insufficient because the magnetic moment of the RE originates from $4f$-electrons lying far below the Fermi level, $E_F$, whereas that of the TM originates from $3d$-electrons across the $E_F$[18-20]. Consequently, an energy-level-dependent approach is required for obtaining a complete understanding of the RE–TM ferrimagnets.

Among various RE–TM ferrimagnets, Tb-based ferrimagnets possess unique characteristics because the $4f$-orbital of Tb lies in the

[1]Department of Physics, Korea Advanced Institute of Science and Technology (KAIST), Daejeon, Republic of Korea. [2]Department of Materials Science and Engineering and KI for Nanocentury, KAIST, Daejeon 34141, Republic of Korea. [3]School of Electrical Engineering and Computer Science, GIST, Gwangju 61005, Republic of Korea. [4]Center for Scientific Instrumentation, KBSI, Daejeon 34133, South Korea. [5]These authors contributed equally: Ji-Ho Park, Won Tae Kim. ✉e-mail: rotermund@kaist.ac.kr; kabjin@kaist.ac.kr

range of a few electronvolts (eV) below $E_F$ that is accessible by visible light[21–23]. In addition, Tb has a strong spin-orbit coupling; the Tb spins are coupled not only to the spins of the TM but also to the lattices, leading to so-called sperimagnetic structures, in which the Tb moments are distributed within a solid angle[24]. Despite these interesting features, Tb-based ferrimagnets has generally been described by the oppositely headed two arrows, in which the magnetic moments of Tb and TM are treated as antiferromagnetically-coupled arrows without considering the energy level[12–17]. An energy-level dependent approach has been proposed for probing the ultrafast spin dynamics in TbFeCo ferrimagnets[21]; however, the moments of TbFeCo was still described by the oppositely headed two arrows. Furthermore, the previous work has only focused on the non-equilibrium regime which occurs in picosecond time scale.

In this work, we investigate the quasi-static natures of perpendicularly magnetized TbCo ferrimagnets using multiple energy-level-dependent approaches (Fig. 1): Hall measurement to probe the magnetic moment near $E_F$, time-resolved magneto-optical Kerr effect (TR-MOKE) and laser-induced terahertz (THz) emission, both of which can access the specific energy level (from $E_F$ to $E_F$ - $E_\lambda$) by varying the wavelength of light $\lambda$, and magnetization measurement using a vibrating sample magnetometer (VSM) that contains the information of the total magnetic moment from the whole energy level ($E_{total}$). The Hall measurement shows that the magnetic moments near the Fermi level hardly respond to the in-plane (IP) magnetic field. However, the VSM indicates that the total moment is rather susceptible to the external IP field. Together, these suggest that the magnetic moments respond differently to the external magnetic field depending on the energy level. The wavelength-selective TR-MOKE reveals that the magnetic moments at deeper energy levels are more susceptible to the magnetic field than those near $E_F$, implying that the direction of magnetic moment at deeper energy level is more easily changed by the magnetic field than the moment near $E_F$. Notably, we found that the magnetic moment at a deeper energy level decays extremely slowly. This suggests that the Tb moment in our TbCo ferrimagnetic alloy exhibits spin-glass-like behavior, possibly due to the random anisotropy of Tb[25–27]. Further experiments on the temperature-dependent THz emission confirm the freezing of magnetic moment with decreasing temperature, another key characteristic of spin-glass system. Our results therefore provide experimental evidence of spin-glass-like characteristics in RE-TM ferrimagnets, which may encourage

further studies not only for spintronic applications but also for spin-glass studies.

## Results

### Hall and VSM measurements on TbCo and Co films

We prepared TbCo ferrimagnetic films consisting of Ta (1.5 nm)/Pt (5 nm)/$Tb_{25}Co_{75}$ (20 nm)/Ta (1.5 nm), as shown in Fig. 2a. We note that our TbCo is Co-dominant state near room temperature ($T = 305$ K) and the Tb-dominant state at low temperature ($T = 20$ K)[15,17] (see Supplementary Note. 1). To compare the characteristics of a ferrimagnet with its conventional ferromagnetic counterpart, we also prepared a ferromagnetic Co film consisting of a Ta (2 nm)/Pt (4 nm)/Co (1.2 nm)/$AlO_x$ (2 nm) layer, as shown in Fig. 2b. The Pt layers in both films were used for investigating the laser-induced THz emission via the inverse spin Hall effect[28,29].

We first performed Hall measurements of TbCo and Co films using the standard four-probe method, as shown in Fig. 2a, b, respectively. We note that Hall measurement indicates the magnetization near $E_F$. The measurement was performed at 305 K. Figure 2c shows the measured Hall voltage of the TbCo film while sweeping the magnetic field along the out-of-plane (OOP) direction. A clear square hysteresis is observed, indicating that the TbCo film exhibits a perpendicular magnetic anisotropy (PMA). The Hall signal corresponds to the anomalous Hall effect (AHE) of TbCo[17,30]; it exhibits clockwise rotation, similar to the ferromagnetic Co film (Fig. 2d). This result suggests that the TbCo alloy is in a Co-dominant state at 305 K. The small voltage steps near $H_{OOP} = \pm 50$ mT in the TbCo film may be attributable to the partial magnetization switching at the film edge because the edge has low coercivity [see Supplementary Note 2 for more discussion]. We also investigate the AHE by applying an IP magnetic field, $H_{IP}$. Here, the direction of $H_{IP}$ was set to be parallel to the current direction to exclude the planar Hall effect. Figure 2e, f show the AHE of TbCo and Co films, respectively, while sweeping $H_{IP}$. The AHE of the TbCo film exhibits a "V"-shaped resistance variation without saturation, whereas that of the Co film exhibits clear saturation behavior around $H_{IP}$ of 600 mT. The variation of the Hall signal for TbCo is extremely small ($\Delta V_H \sim 40$ μV for $H_{IP} = 600$ mT). This is only 4% of the total Hall signal change from perpendicular to IP (1 mV, see Fig. 2c), suggesting that the magnetic moment of TbCo near $E_F$ is hardly tilted along the IP direction, whereas that of Co tilts completely along the IP direction by 600mT of $H_{IP}$.

Figures 2g–j show the field-dependent magnetization variation measured by VSM reflecting magnetization from the whole energy level. Details about VSM data processing are shown in Supplementary Note 3. Clear square hysteresis similar to that of Hall measurement was observed upon sweeping $H_{OOP}$ for both TbCo (Fig. 2g) and Co (Fig. 2h), confirming the PMA of both samples. The $H_{IP}$ dependence indicates that the magnetization from the whole energy level of TbCo increases without saturation (Fig. 2i), whereas that of Co is saturated around $H_{IP} = 600$mT, similar to the Hall measurement (Fig. 2j). Interestingly, however, the magnitude of magnetization variation for TbCo obtained by VSM measurement exhibits an unexpected behavior. The total magnetization variation caused by $H_{IP}$ (Fig. 2i) is comparable to that caused by $H_{OOP}$ (Fig. 2g) and shows clear differences compared to the Hall measurement, where the Hall signal variation caused by $H_{IP}$ is only 4% of that caused by $H_{OOP}$ (see Fig. 2c, e). The clear difference between Hall and VSM measurements observed exclusively in TbCo suggests that the magnetization variation of TbCo may sensitively depend on the energy level because the Hall signal originates from the magnetic moment near $E_F$, whereas the VSM measures the magnetic moment of the whole energy level (see Fig. 1). Therefore, the large variation by $H_{IP}$ in the VSM measurement compared to the Hall measurement implies that the magnetic moment at a deeper energy level may be more easily altered by the magnetic field than that near the $E_F$. This unexpected result calls for more advanced measurement techniques that can

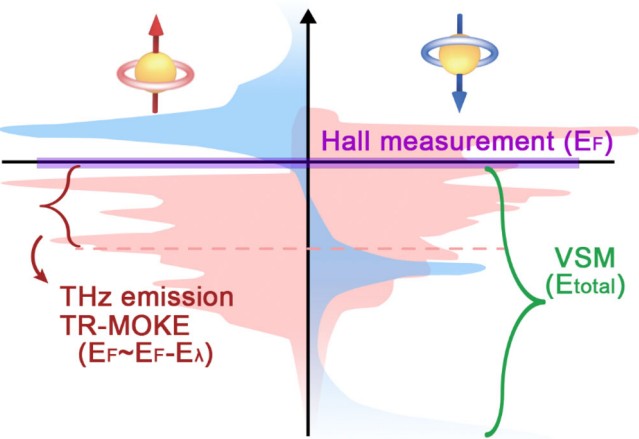

**Fig. 1 | Schematic of spin-dependent band diagram of TbCo with accessible energy levels by different measurement techniques.** Red and blue colors denote the band diagram of the Co 3$d$-shell and the Tb 4$f$-shell, respectively, which was reproduced from ref. 18 (10.1103/PhysRevB.88.104415). The covered energy levels for different measurement techniques are shown in the band diagram.

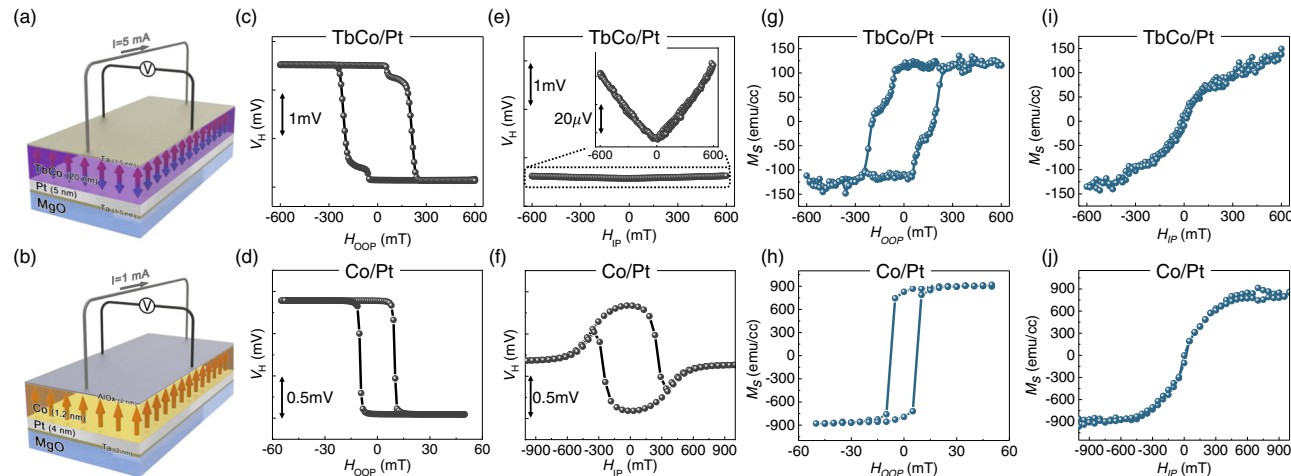

**Fig. 2 | Hall and VSM measurement results. a, b** Schematic of electrical Hall measurement for TbCo and Co. **c, d** Hall voltage as a function of OOP magnetic field, $H_{OOP}$, for TbCo and Co. **e, f** Hall voltage as a function of IP magnetic field, $H_{IP}$, for TbCo and Co. **g, h** Magnetization with respect to $H_{OOP}$ for TbCo and Co. **i, j** Magnetization with respect to $H_{IP}$ for TbCo and Co. All measurement results were obtained at room temperature ($T = 305$ K).

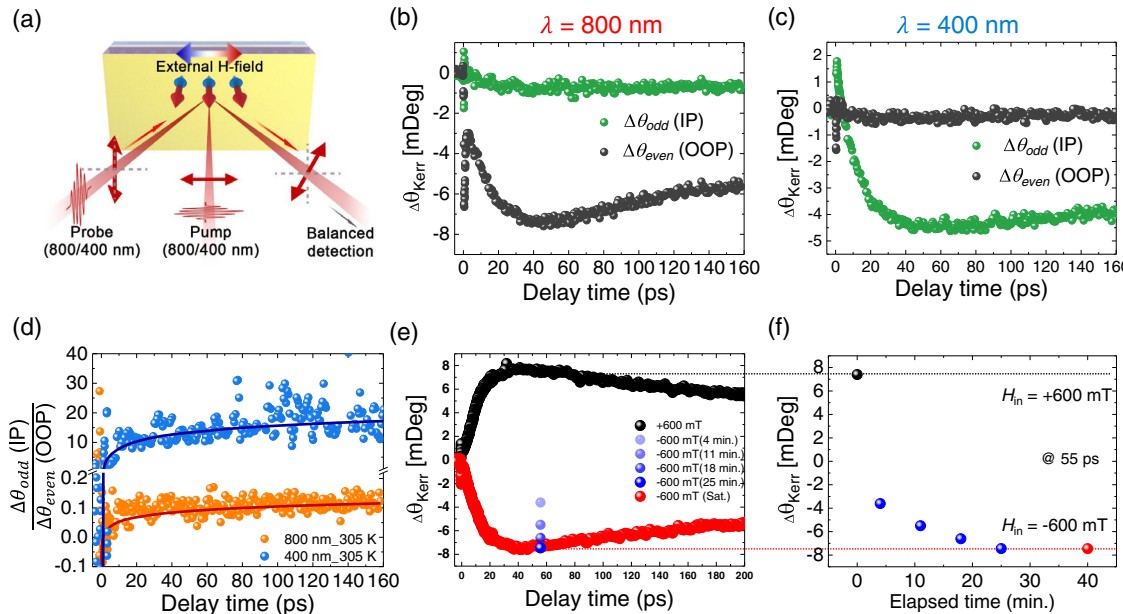

**Fig. 3 | TR-MOKE measurement results for TbCo. a** Schematic of TR-MOKE experiment. **b, c** $\Delta\theta_{even}$ (gray) and $\Delta\theta_{odd}$ (green) at (**b**) $\lambda = 800$ nm and (**c**) $\lambda = 400$ nm at 305 K. Measurements were performed after applying a magnetic field of $H_{IP} = 500$ mT and confirming that the Kerr signal is stabilized. **d** Ratio of $\Delta\theta_{odd}$ and $\Delta\theta_{even}$ depending on wavelength. **e** The 400 nm pump-probe raw data under external magnetic fields ($\pm 600$ mT) at 305 K. **f** Slow change of Kerr signal at fixed delay time of 55 ps when the external magnetic field is changed from +600 mT to −600 mT.

access the energy level between $E_F$ and specific energy level. We investigate this using optical measurements (e.g., TR-MOKE and THz emission) by changing the wavelength of light $\lambda$.

## TR-MOKE measurements on TbCo film

Figure 3a illustrates the TR-MOKE measurement. The sample was excited by optical pump pulses with a normal incidence angle to the sample surface, and the probe beam was injected at the maximum incident angle of 30° with respect to the normal in our setup. The oblique incident angle enables us to detect both OOP and IP Kerr signals, the even component for $H_{IP} - \Delta\theta_{even} = [\Delta\theta_K(+H_{IP}) + \Delta\theta_K(-H_{IP})]/2$ —corresponds to the Kerr signal from OOP magnetic moment, and the odd component for $H_{IP} - \Delta\theta_{odd} = [\Delta\theta_K(+H_{IP}) - \Delta\theta_K(-H_{IP})]/2$ —corresponds to that from IP magnetic moment. The pulses wavelength of

$\lambda = 800$ nm and $\lambda = 400$ nm generated by second-harmonic generation were used for both the pump and the probe beams. This enabled accessing two distinct energy levels: 1.55 eV for $\lambda = 800$ nm and 3.1 eV for $\lambda = 400$ nm. The TR-MOKE setup is described in detail in the "Methods" section.

Figure 3b shows the transient Kerr signal variation of the TbCo film after exciting and probing the sample using ultrashort pulses at 800 nm. The $\Delta\theta_{even}$ (black) and $\Delta\theta_{odd}$ (green) corresponding to the OOP and IP Kerr signal variations, respectively, were measured as a function of delayed time. Here, we used $H_{IP} = \pm 500$ mT and performed the measurement at 305 K. The laser-induced demagnetization profile is clearly observed in $\Delta\theta_{even}$ (OOP signal); the Kerr signal initially drops rapidly, followed by slow demagnetization before it recovers[31]. However, the variation of $\Delta\theta_{odd}$ (IP signal) is extremely small, indicating

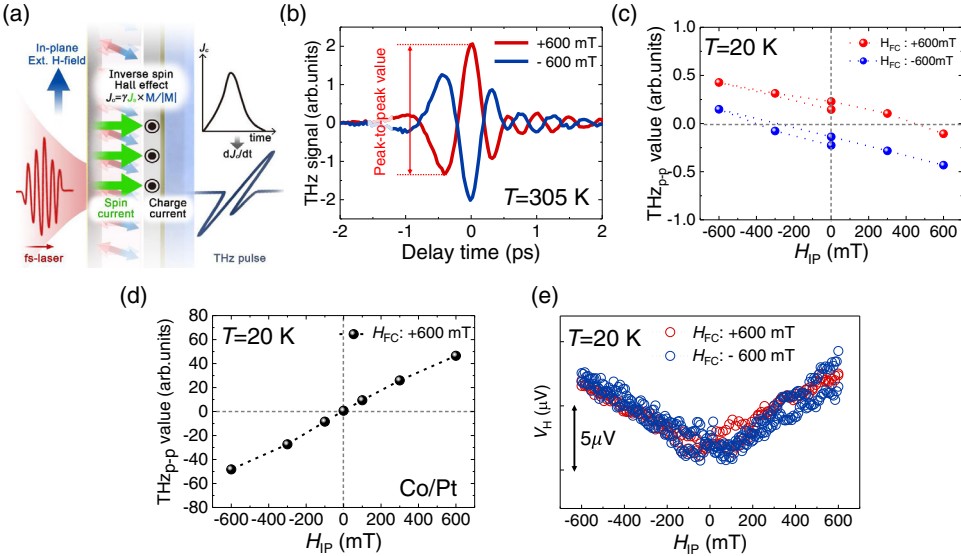

**Fig. 4 | THz emission measurement results. a** Schematic of principle of THz emission in magnetic/nonmagnetic bilayer sample. **b** Measured THz time traces according to external magnetic fields with 400-nm pump at 305 K. **c** Peak-to-peak values of THz waves generated from TbCo/Pt according to field-cooling conditions with 400-nm pump at 20 K ($H_{FC}$ = ± 600 mT). **d** Peak-to-peak values of THz waves generated from Co/Pt with 400 nm pump at 20 K ($H_{FC}$ = 600 mT). **e** IP Hall measurement results at 20 K ($H_{FC}$ = ± 600 mT).

that in the energy level of $E_F−E_\lambda$ = 800 nm (0–1.55 eV), the demagnetization is dominantly induced by the OOP magnetic moment under $H_{IP}$ = ±500 mT. Interestingly, however, the trend was reversed when the same experiment was conducted at the pump/probe wavelength of 400 nm, as shown in Fig. 3c. At this wavelength, the demagnetization profile is more pronounced in $\Delta\theta_{odd}$ (IP signal) than in $\Delta\theta_{even}$ (OOP signal). For a clear comparison, Fig. 3d shows the ratio between IP and OOP Kerr signals obtained at $\lambda$ = 800 nm (orange) and $\lambda$ = 400 nm (blue). The IP Kerr signal is clearly more dominant than the OOP Kerr signal at $\lambda$ = 400 nm, indicating that the magnetic moments at a deeper energy level are more susceptible to $H_{IP}$ (see Supplementary Note 4 for more quantitative approach). This distinct response to the magnetic field explains why the Hall and VSM measurements show different trends for $H_{IP}$, as shown in Fig. 2e, i. (further comparability check between optical measurement and Hall and VSM measurement is discussed in Supplementary Note 5). Notably, the energy levels of 1.55 eV (800 nm) and 3.1 eV (400 nm) do not directly correspond to the Co 3$d$-band and Tb 4$f$-band, respectively, in our TbCo sample because the Co 3$d$-band lies across the $E_F$ and the maximum density of state of the Tb 4$f$-band is likely to be lower than 3.1 eV [see Supplementary Note 6]. In this regard, our optical technique is not an element-specific method[21]; rather, it is an energy-level-selective detection technique. Nevertheless, we believe that the light at $\lambda$ = 400 nm is more closely accessible to the Tb 4$f$-band than that at $\lambda$ = 800 nm. Accordingly, the results obtained using the 400-nm pump/probe beams reflect the characteristics of the moment from Tb sub-lattice more clearly.

Notably, we observed that the temporal response of the Tb magnetic moment under changing $H_{IP}$ was remarkably slow. The black data in Fig. 3e represents the laser-induced ultrafast demagnetization and recovery profile, which was measured by 400 nm pump/400 nm probe TR-MOKE. Here, we applied in-plane magnetic field of +600 mT and waited sufficiently long time for the magnetic moment direction to be stabilized. After measuring the data, we reversed the magnetic field to the opposite direction, i.e., H = −600 mT, and then measured the Kerr signal at $t$ = 4 min, 11 min, 18 min, 25 min after applying the magnetic field. This is because the Kerr signal does not reverse instantaneously, but takes long time to be fully saturated. Here, we only monitored the Kerr signal at a fixed delay time of 55 ps (blue symbols in Fig. 3e), because the full profile measurement takes time and the

Kerr signal gradually changes even during the measurement. After $t$ = 40 min, we confirmed that the Kerr signal does not change anymore, so we measured the full Kerr signal profile (red data in Fig. 3e). Figure 3f shows the gradual change of Kerr signal in time (which corresponds the symbols in Fig. 3e), highlighting that the response of the magnetic moment to the magnetic field does not occur immediately but takes a relatively long time of the order of tens of minutes. We note that we also checked the slow response of magnetic moment using quasi-static measurement (see Supplementary Note 7). This interesting feature has not been reported thus far.

The slow spin dynamics is a typical feature of spin-glass systems[32]. Therefore, what we observed in TbCo may be related to the spin-glass characteristics. The spin-glass is a system where multiple ground states exist. In this case, the thermal hopping between the ground states, which typically takes a long time, can occur. This should be the origin of slow dynamics. Furthermore, such spin-glass characteristic can be verified by lowering temperature, because the thermal hopping is strongly suppressed at low temperature and thus, the moment freezing can be observed. To confirm this, we further studied the magnetic moment freezing with decreasing temperature by using THz emission experiment.

## THz emission measurements on TbCo and Co films
THz emission experiments on TbCo film were performed to verify the spin-glass-like freezing because the THz emission signal is generally proportional to the IP magnetization component before pump pulse incidence[29]. Figure 4a illustrates the THz emission principle in a magnetic/nonmagnetic bilayer sample. A fs-laser pulse at $\lambda$ = 400 nm is injected to the normal direction on the sample surface to induce transient spin current flowing into the neighboring nonmagnetic metal[33,34]. The spin current is converted into the transverse IP charge current through the inverse spin Hall effect (ISHE), thereby generating THz waves [see Supplementary Note 8]. In this geometry, the IP magnetization component of the magnetic layer influences the THz emission. Figure 4b shows the measured THz signal of the TbCo film for $H_{IP}$ = + 600 mT (red) and $H_{IP}$ = −600 mT (blue) at 305 K. The reversed polarity for the opposite $H_{IP}$ indicates that the THz signal is indeed linked to the IP magnetization [see Supplementary Note 9 for further confirmation].

Figure 4c shows the THz peak-to-peak amplitude as a function of $H_{IP}$ measured at $T = 20$ K after field cooling with $H_{FC} = +600$ mT (red) and $H_{FC} = -600$ mT (blue) [see Supplementary Note 10 for raw THz signal]. Here, field cooling was performed by reducing the temperature from $T = 305$ K to $T = 20$ K while applying the IP field of $H_{FC}$. The zero-crossing point clearly shifts to the positive or negative field direction (the amount of shift is about 400 mT) depending on the sign of $H_{FC}$ in TbCo, indicating that the magnetic moment is frozen during the field-cooling process. To identify the origin of this spin-glass-like freezing, we measured the THz emission of the Co film after the same field-cooling procedure. As shown in Fig. 4d, the Co sample does not show any shift with field cooling, suggesting that the spin-glass-like freezing is inherently owing to the existence of the Tb moment in the TbCo ferrimagnet. To further confirm the energy-level dependence of spin-glass-like freezing, we measured the Hall voltage after the same field-cooling procedure. Figure 4e shows the Hall resistance variation under the sweeping of $H_{IP}$. A "V"-shaped Hall signal similar to that in Fig. 2e was observed; however, an overall field-shift as large as 400 mT (see Fig. 4c) was not observed in the Hall measurement. This indicates that the magnetic moment near the $E_F$ does not show spin-glass-like freezing. Consequently, the Hall measurement confirms that the spin-glass-like characteristics can only originate from the magnetic moment at a deeper energy level where the Tb moment is dominant.

## Discussion

We further discuss the possible physical origin of the spin-glass-like properties of TbCo. Since there is no universal Hamiltonian to describe the spin glass[32], the origin of spin-glass has been studied individually according to the specific system. For example, spin glass characteristic of $Eu_xSr_{1-x}S$ was explained based on dipolar interaction[35] and that of CuMn was described based on DM (Dzyaloshinskii–Moriya) type interaction[36]. The fundamental reason of spin-glass-like properties of TbCo is not perfectly understood at the present stage and requires further studies. However, we think that it is possibly due to the strong random anisotropy of Tb. According to ref. 37, the correlated spin glass state can be achieved under the condition for $H_{applied} < \frac{H_r^4}{H_{ex}^3}$, where $H_{applied}$ is the applied field, $H_r$ is the random anisotropy field and $H_{ex}$ is the exchange field. This implies that the spin-glass properties can appear when the $H_r$ is bigger than $\left(H_{applied}H_{ex}^3\right)^{1/4}$. With the experimental value of $H_{applied} = 600 mT$, and literature value of $H_{ex} = 138 T$ for Tb at $300K$[38], we can roughly estimate the spin-glass condition as $H_r > 35.4 T$, which might be the case of our TbCo (see Supplementary Note 11 for $H_r$ measurement).

Lastly, we discuss the importance of our results. Given that the spin-glass-like properties arises from the characteristics of Tb, we infer that our finding is a generic phenomenon in various RE-based ferrimagnets because RE materials generally share similar properties [see Supplementary Note 12 for further demonstration]. We also expect that our findings crucially affect the study of ferrimagnetic spin dynamics. For example, the spin-orbit-torque switching efficiency of ferrimagnets or the compensation point of ferrimagnets is a focus of interest, but errors may be included when the in-plane magnetic field is applied, because two moments in ferrimagnets respond differently to the in-plane magnetic field. Furthermore, spin-glass-like slow dynamics of ferrimagnet suggests that one should be very careful when they saturate the magnetization of ferrimagnets, because it has multiple ground states and takes very long time for the saturation (min to hours). In addition, our finding that the ferrimagnetic TbCo exhibits spin-glass-like properties even at room temperature implies that the ferrimagnet could be useful even in the spin-glass study.

To summarize, we investigated the field-response of the magnetic moment of ferrimagnetic TbCo using multiple techniques such as Hall measurement, VSM measurement, TR-MOKE, and THz emission measurements to access different energy levels. We revealed that the

magnetic moments of TbCo exhibit distinct responses to the magnetic field depending on the energy level. The magnetic moment near $E_F$ is slightly tilted by the IP magnetic field, and that lying far below the $E_F$ is easily altered by the IP magnetic field. Furthermore, we found that the magnetic moment at a deeper energy level exhibit spin-glass-like characteristics such as slow relaxation and magnetic moment freezing under field cooling. These features are attributed to the properties of Tb, including random ion anisotropy and distinct $4f$ energy level, that are generic characteristics of REs. Further theoretical studies are required to understand the microscopic details of distinct characteristics of TbCo. Therefore, our results are expected to encourage further studies of various RE–TM ferrimagnets in consideration of the energy-level-dependent spin dynamics.

## Methods
### Sample preparation
We prepared TbCo ferrimagnetic films by DC magnetron sputtering. The films consist of Ta (1.5 nm)/Pt (5 nm)/$Tb_{25}Co_{75}$ (20 nm)/Ta (1.5 nm) layers on an MgO substrate, as shown in Fig. 2a. The bottom and top Ta layers served to enhance adhesion and prevent oxidation, respectively. The Pt layer was used for investigating the laser-induced THz emission via the inverse spin Hall effect[28,29]. The Tb:Co composition ratio was determined to verify the Co-dominant state near room temperature ($T = 305$ K) and the Tb-dominant state at low temperature ($T = 20$ K)[15,17] (see Supplementary Note 1). To compare the characteristics of a ferrimagnet with its conventional ferromagnetic counterpart, we also prepared a ferromagnetic Co film consisting of a Ta (2 nm)/Pt (4 nm)/ Co (1.2 nm)/$AlO_x$ (2 nm) layer on the same substrate, as shown in Fig. 2b.

### Hall and VSM measurements
Hall measurements were conducted with a standard measurement geometry, as shown in Fig. 2a, b. The room-temperature Hall measurement was performed on a homemade probe station that was equipped with a rotating stage and an electromagnet (GEM 120, RNDWARE), and the low-temperature Hall measurement was performed using a Physical Property Measurement System (PPMS-9, Quantum Design). The OOP and IP directions were precisely determined by rotating the sample stage with an accuracy of 1°. The IP magnetic field was applied along the current direction to avoid possible artefacts from the planar Hall effect. Additionally, the magnetization was determined using a VSM (LakeShore 7400). The linear background signal originating from the substrate was subtracted by independent measurements of the signal from the bare substrate.

### THz emission and TR-MOKE experiments
**A1. Experimental setup.** For investigating and comparing the characteristics of ferrimagnet TbCo films depending on external conditions such as temperature, optical pump and probe wavelengths, and external magnetic field, we conducted THz emission spectroscopy (TES) and TR-MOKE experiments. Figure S14 shows the experimental setups for TES, static polar MOKE, and TR-MOKE. The sample was mounted on a cooper holder at the center of a vacuum chamber (Cryostation s50-MO, Montana Instruments). A fused silica window with antireflection coating for a broad wavelength range of 400–1000 nm was used to input the optical pulses, and a TPX window that provides relatively high transmission of THz waves was used to output the THz emission. The cryogenic system can control the temperature of the copper holder from 4 K to 340 K and sweep the external magnetic field from −600 mT to +600 mT through bipolar magnets in a direction parallel to the sample surface. We used optical pulses with two different wavelengths for static MOKE as well as to pump and probe the TbCo film and compare energy-level-dependent responses. Near-infrared pulses from a 1-kHz Ti:sapphire regenerative amplifier system operating at 800 nm (Spitfire Ace, Spectra Physics)

and ultraviolet pulses at 400 nm generated by second-harmonic generation were used for the investigation. The pulse duration and polarization state of pulses at two different wavelengths were approximately 130 fs and horizontal (pump) at the sample position, respectively. Note that the optical pulses were spilt into three beams: an intense pump beam was used to stimulate the magnetization of the TbCo film, a TR-MOKE probe beam was spatiotemporally synchronized with respect to the pump beam at the sample position to detect the time-dependent magnetization change through MOKE measurements, and a sampling beam was used to record the time trace of the THz electric fields through the electro-optic (EO) sampling method. The radius and fluence of the pump beam were approximately 0.7 mm and 0.71 mJ/cm$^2$, respectively.

The sample was excited by the pump beam at a normal incidence angle to the sample surface. After passing the sample, the transmitted residual pump was blocked by a Si-wafer because this residual pump can distort the EO-sampling signal during the detection of THz waves. To detect the THz waves emitted from the sample, a pair of 90° off-axis parabolic mirrors was used to guide and focus onto the EO-detection crystal (a 2 mm-thick <100> ZnTe). All THz emission experiments were conducted under conditions of low humidity with dry air purging to reduce the water vapor absorption of THz waves.

To measure the IP magnetic moments of the sample (IP MOKE), the incident angle of the MOKE probe beam was kept at the maximum angle of 30° that is available in our experimental setup. The incident polarization and beam size of the probe beam were vertical and ~70 μm, respectively.

**A2. TR-MOKE.** The MOKE probe beam reflected from the sample was detected using the balanced detection method after passing through a half-wave plate and Wollaston prism. The intensity difference of the balanced detector, which was proportional to the Kerr signal, was initially set to zero by rotating the half-wave plate. Based on the relation between the intensity difference values obtained by rotating the half-wave plate and by the Kerr effect, we could estimate the Kerr signal from the obtained signal. In our experimental configuration, the Kerr signals were influenced by changes in both the OOP and IP magnetization owing to the oblique incidence and OOP anisotropy of the sample. Because the Kerr signal was affected by the linear sum of each magnetic direction[39], the signal for each direction could be obtained by subtracting the measured signal according to the applied IP magnetic fields of the same strength in the opposite direction. If $\Delta\theta_K$ ($\pm$500) was the measured Kerr signal when an IP magnetic field of $\pm$500 mT was applied, $\Delta\theta_K$ from OOP and IP magnetic moments were obtained as [$\Delta\theta_K$(+500 mT) $\pm$ $\Delta\theta_K$(−500 mT)]/2. All experiments were performed with a nearly identical pump power, beam size, incident angle, and temperature at wavelengths of 400 and 800 nm.

**A3. THz wave emission.** In heterostructures consisting of magnetic and nonmagnetic (NM) materials with strong spin-orbit coupling, ultrashort pulses can induce transient spin-polarized currents flowing into the NM layer through super diffusion and spin-dependent Seebeck effect[40,41]. This transient spin current is converted into a transverse IP charge current through ISHE. The converted transient charge current subsequently generates THz waves that are perpendicular to the magnetization and propagation of the spin current and are expressed by

$$J_C^{\perp} = \gamma \mathbf{J_s} \times \mathbf{M}/|\mathbf{M}| \tag{1}$$

where $J_C^{\perp}$ is the converted transient charge current; $\gamma$, the spin Hall angle; $\mathbf{M}$, the magnetization; and $\mathbf{J_s}$, the injected spin current[41]. Thus, THz waves generated by ISHE are influenced by the IP magnetization of the magnetic layer. For common magnetic materials possessing an OOP magnetic moment, the strength and polarity of the generated THz electric fields change depending on the strength and direction of the IP external magnetic field, as shown in Fig. 4d. For investigating the field-cooling effect, the sample was cooled down from 305 K to 20 K at applied IP external magnetic fields of ± 600 mT. The THz magnitudes were confirmed to be substantially different at ±600 mT with the field-cooling condition than at 305 K (comparison of Figs. S11a, b with Fig. 4b). In contrast, for the ferromagnetic Co/Pt film, the polarity of the THz waves was not influenced by the temperature and field-cooling condition, as shown in Fig. S11c and Fig. 4d. Note that the THz peak-to-peak values shown in Fig. 4(c) and (d) include the polarity information, where the peak-to-peak value of the generated THz electric field with IP magnetic field of +600 mT and −600 mT at 305 K was set as positive and negative, respectively.

## Data availability
The datasets generated during this study are available from the corresponding authors on request.

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

## Acknowledgements
This research was supported by the National Research Foundation of Korea (NRF) funded by the Korean Government (MSIP) [grant numbers: 2020R1A2C4001789, 2016R1A5A1008184, 2019R1A2C3003504] and KAIST-funded Global Singularity Research Program for 2021. Y.J. was supported by the Institute of Information & communications Technology Planning & Evaluation (IITP) grant funded by the Korea government (MSIT) (No.2022-0-01021). BSH also acknowledged the support by the ICT R&D program of MSIT/IITP (No.2021-0-01810), the development of elemental technologies for an ultrasecure quantum internet.

## Author contributions
J.-H.P. and W.T.K. planned the research under the supervision of K.-J.K. and F.R. J.-H.P. performed the Hall and VSM measurements and W.T.K. performed the TR-MOKE and THz emission measurements with the help of B.S.H., and they analyzed the results. W.W. assisted with the Hall measurements. J.-H.K., S.L., and B.-G.P. performed the deposition of TbCo and Co/Pt films. Y.J. performed MPMS measurement. J.-H.P., W.T.K., F.R., and K.-J.K. wrote the manuscript. All authors were involved in the discussion of the results and commented on the manuscript.

## Competing interests
The authors declare no competing interests.
