## [Peer Review File · Nature Communications]

Reviewers' Comments:

Reviewer #1:

Remarks to the Author:

The work by Park and company presents an experimental study of primarily TbCo/Pt, but also Co/Pt samples, where a range of techniques have been used to investigate various static and dynamic properties. Several features of the IP and OOP hysteresis loops measured using Hall and VSM measurements are discussed in the context of the magnetic moments arising from different band electrons (at different energies) having different response. Overall, the experimental work appears rigorous and well carried out, but overall, I don't find the findings particularly surprising. There have been previous works investigating the wavelength-dependent studies of ultrafast spin dynamics (Khorsand et al. PRL vol 110, 107205 (2013)), which the authors cite in the context of element specificity. However, this is also essentially an energy dependent probe. Also, the work of Frietsch et al. (Nat Commun, vol 6, 8262 (2015)) investigates the dynamics of the moments arising from different bands. Furthermore, whilst there is a nice set of different experimental methods presented here, the methods in themselves are not new. The article is more magnetism focused and the results not of interest to a broad readership of Nature Communications, therefore I would not recommend it for publication.

I also have specific several comments on the manuscript that I feel would improve the article.

1. The results of figure 2 are of hysteresis (quasi-static) loops and those of figure 3 non-equilibrium. The responses of the moments from the different bands will be different depending on the time-scale, so I am not convinced by the comparability between the two sets of results.
2. I found that referring to the work as "energy dependent" a little (I think unintentionally) misleading. From the title I had the impression that the measurements would be probing something across a spectrum of energies, rather than just a select few.
3. I could not follow what exactly was done in figure 2e. The figure is unclear and the description in the text I found didn't really help. There is talk of switching not being instantaneous, which, again, I could not follow. Why would one expect a switching process through the application of a field to be instantaneous? Why did it take 10 minutes? What is the process by which this occurred? More detail needs to be provided here.
4. Also related to point 3. The discussion around the spin-glass characteristics was also not clear. There is no real introduction to the reader about spin-glasses, which is rather niche. The presence of the spin-glasses is introduced to explain the slow "switching" behaviour, which, as mentioned in point 3 is not clear.

Minor Points

1. The authors refer to TbCo as being antiferromagnetic, e.g. line 32/33, "As a result, the antiferromagnetic spin configuration". It does not have an antiferromagnetic spin configuration, but a ferrimagnetic with two sublattices with antiferromagnetic exchange coupling. This point should be made clearer throughout.
2. The reference to a "submoment" is not clear.
3. In figure 3 or its caption, mention which material is being studied.

Reviewer #2:

Remarks to the Author:

Ji-Ho Park et al investigated the magnetic moment configuration of ferrimagnetic TbFe alloys using various methods including Hall effect, VSM, TR-MOKE, and THz emission, and distinguished the sub-moments of Co3d and Tb4f under pump/probe light with wavelength of 800 and 400 nm, respectively. This method seems to be an effective way, but the following issues should be addressed first before publication.

1. For TR-MOKE, the authors say that the IP Kerr signal at 400 nm is more dominant than the OPP one, indicating the magnetic moments at a deeper energy level are more susceptible to HIP. What

is the relation of the IP/OPP Kerr signal with the magnetic moments at fermi/deeper energy level?

2. In caption of Figure 3, with an in-plane magnetic field $H_{IP} = 500$ mT, confirming the saturation of the magnetic moment. Please see the VSM curve in Fig.2(i), it shows not saturated even at $H_{IP} = 600$ mT. In fact on the contrary, if saturated, the magnetic moment is totally in the film plane, then the OPP Kerr signal should not be detectable.
3. The whole article should make clear the description of Kerr rotation/angle with the Kerr signal. They are two different parameters.
4. About the figures of 3(b) and 3(c), the changes of Kerr angle at the beginning are not clear. They need enlarged insets with smaller delay time within several picoseconds.

Reviewer #3:

Remarks to the Author:

In this manuscript, the authors have studied and compared the magnetic properties of ferrimagnetic TbCo alloy with ferromagnetic Co films by the AHE, VSM, TR-MOKE, and THz emission measurements. They found the variation of the Hall signal for TbCo is extremely small, only 4% of the total Hall signal change from perpendicular (HOOP) to in-plane (HIP) field of 600 mT. In contrast, the VSM loops indicate that the total magnetization variation caused by HIP is comparable to that caused by HOOP. Based on the difference between the in-plane Hall and VSM results, the authors conclude that the magnetic moments respond differently to the external magnetic depending on the energy level of 3d and 4f, and the Tb submoment at a deeper energy level is more easily changed by the magnetic field than the Co submoment near the EF. The reviewer has some concerns about these opinions. Firstly, we know the AF coupling in TbCo is very strong, so the Co magnetic moments should rotate simultaneously with the magnetic moments of Tb. It is not possible to keep Co undisturbed when Tb is rotated by applying HIP in the range of just ± 600 mT. Secondly, the observed similar magnetization changes of IP and OOP VSM loops for the TbCo sample are not reasonable, since the HOOP loop has been already saturated while the HIP loop seems far from saturation in Fig.2. The unsaturated IP loop suggests that the magnetization variation should be much smaller than the OOP loop. I am afraid that the small net magnetization of TbCo is hardly tilted by HIP and the obtained similar value in Fig. 2 may probably arise from the background signal which is not correctly subtracted from the raw data. Lastly, for the control sample of Co film, the Hall and VSM results are very consistent, both can be switched by applying an IP field of ± 600 mT, suggesting that the Co moments are actually not difficult to be changed. According to my knowledge, whether the magnetization orientation can be changed or not, relies on the total net magnetization, the effective magnetic anisotropy field as well as the applied field. Therefore, I could not recommend its publication in Nature Communication unless the authors can provide more convincing evidences.

Reviewer #1 (Remarks to the Author):

The work by Park and company presents an experimental study of primarily TbCo/Pt, but also Co/Pt samples, where a range of techniques have been used to investigate various static and dynamic properties. Several features of the IP and OOP hysteresis loops measured using Hall and VSM measurements are discussed in the context of the magnetic moments arising from different band electrons (at different energies) having different response. Overall, the experimental work appears rigorous and well carried out, but overall, I don't find the findings particularly surprising. There have been previous works investigating the wavelength-dependent studies of ultrafast spin dynamics (Khorsand et al. PRL vol 110, 107205 (2013)), which the authors cite in the context of element specificity. However, this is also essentially an energy dependent probe. Also, the work of Frietsch et al. (Nat Commun, vol 6, 8262 (2015)) investigates the dynamics of the moments arising from different bands.

→ We thank the reviewer for the valuable comment and for letting us remind the important literatures. As the reviewer mentioned, our work might be seen as similar to previous reports in that both works discuss the energy-dependent probe. However, there is a substantial difference between our work and previous works, as we will describe in the following.

Khorsand *et al.* reported the *non-equilibrium nature* of ferrimagnet which occurs at *ps* time scale. The main findings of their work are two folds: 1) Tb and FeCo temporarily align ferromagnetically at the *ps* time scale, and 2) the demagnetization profile of Tb is different from that of FeCo. We also would like to note that their results were analyzed based on the *rigid two-arrow model* where the magnetic moments of Tb and FeCo are treated as antiferromagnetically coupled rigid arrows [Fig. 3 of their paper]. Unlike their work, we investigated the *quasi-static nature* of ferrimagnet. We found that the Tb and Co respond differently to the external in-plane magnetic field, which highlights that *the moments of Tb and Co are not rigidly bound*, but rather freely respond to some extent. This leads to the canting of Tb and Co moments, which cannot be interpreted based on rigid two-arrow model. Furthermore, we also discovered that *the ferrimagnetic TbCo exhibits the spin-glass-like slow dynamics, which occurs at quasi-static time scale (min ~ hours)*. This spin-glass-like property of ferrimagnets is very surprising, as it occurs at room temperature and leads to the freezing of magnetic moment at low temperature. Therefore, our work claims different physics and phenomena from that of Khorsand *et al.* We note that the work by Frietsch *et al.*, also discussed the non-equilibrium dynamics of itinerant and localized magnetic moment of single Gadolinium, which occurs at *ps* time scale.

We think that the wavelength-dependent ultrafast demagnetization profile observed by TR (time resolved)-MOKE in Fig. 3 may lead the reviewer to think that our work is similar to the previous works.

To clear up the misunderstanding, we provide *quasi-static* measurement results in the revised version (see the response to the comment 1 below), which corroborates the distinct *quasi-static nature* of TbCo.

Furthermore, whilst there is a nice set of different experimental methods presented here, the methods in themselves are not new. The article is more magnetism focused and the results not of interest to a broad readership of Nature Communications, therefore I would not recommend it for publication.

→ We agree with the reviewer that the experimental methods presented here are not new. However, we believe that we provide a new perspective of the combination of different measurement techniques. In this study, we used various techniques, such as VSM and Hall measurement, TR- and static-MOKE, THz emission, which are widely used in the community. In particular, the VSM and Hall measurement are the conventional quasi-static measurement techniques which have been used in various studies to characterize the magnetic properties of materials, as they provide the magnetic hysteresis loop. In the previous studies, both techniques have equivalently been considered to represent the magnetic moment of materials. However, here we found that two techniques give different hysteresis shapes, especially when we measure the ferrimagnets. This highlights that the energy-dependent characteristics of measurement techniques must be considered when we study the ferrimagnets. As the ferrimagnets become the center of research interest [see the review paper by SK Kim *et al.*, *Nat. Mater.* **21**, 24 (2022)], our finding could provide an important message to the community that the *energy-level dependence of measurement technique must be considered even in conventional techniques such as VSM and Hall measurements*.

We also remark the broad interest of our work. Ferrimagnets become one of the most important magnetic materials in condensed matter physics [SK Kim *et al.*, *Nat. Mater.* **21**, 24 (2022)]. This is because the ferrimagnets provide a promising platform to realize the antiferromagnetic spintronics where the ultimate spintronic device is expected to be made [see the review papers, e.g., V. Baltz *et al.*, *Rev. Mod. Phys.* **90**, 015005 (2018), T. Jungwirth *et al.*, *Nat. Nanotech.* **11**, 231 (2016), O. Gomonay *et al.*, *Nat. Phys.* **14**, 213 (2018)]. Indeed, various antiferromagnetic properties, such as fast spin dynamics, high frequency operation, long spin coherence, and the field-immunity have been secured in ferrimagnets [see *Nat. Mater.* **21**, 24 (2022) and references therein]. Despite the advances, however, most of studies so far have described the ferrimagnets using *antiferromagnetically-coupled rigid two-arrow model* where the rare-earth (e.g., Tb) and transition metal (e.g., Co) are treated equivalently. In our work, we found that *this simple model is not sufficient to describe the ferrimagnets even at quasi-static regime*, because two moments in ferrimagnets lie at different energy bands, and respond differently to the external magnetic field, which leads to the different magnetic response depending on

the measurement techniques. We also revealed that *the ferrimagnets exhibit spin-glass-like slow dynamics even at room temperature*. These findings are important and should urgently be known to the community, because one can lead to incorrect results if one does not consider the spin-glass-like properties of ferrimagnets. For example, the spin-orbit-torque switching efficiency of ferrimagnets or the compensation point of ferrimagnets is a focus of interest, but errors may be included when the in-plane magnetic field is applied, because two moments in ferrimagnets respond differently to the in-plane magnetic field. Furthermore, spin-glass-like slow dynamics of ferrimagnet suggests that one should be very careful when they saturate the magnetization of ferrimagnets, because it has multiple ground states and takes very long time for the saturation (min to hours). Therefore, we believe that our work provides an urgent message to the community, and thus could impact on the emerging field of ferrimagnetic spintronics as well as antiferromagnetic spintronics.

We also note that our work could contribute to the field of spin-glass research. So far, the representative platform of spin glass has been magnetic nanoparticles (NPs) and single molecule magnets (SMMs). In these systems, interparticle interaction, random orientation of unidirectional anisotropy, inherent disorder, and frustration lead to a competition of different spin alignments, resulting in spin-glass characteristics such as multiple ground states and slow dynamics. These spin-glass characteristics can be connected to magnetic memory effect or quantum computing [M. Mannini *et al.*, *Nat. Mater.* **8**, 194 (2009), G. Christou *et al.*, *MRS Bulletin* **25**, 66 (2000)]. However, spin-glass materials are usually working at low temperature [M. Ali *et al.*, *Nat. Mater.* **6**, 70 (2007)], which prevents further developments. In this study, we found that the ferrimagnetic TbCo exhibits spin-glass-like slow dynamics even at room temperature. This suggests that *the ferrimagnet can be another candidate for the spin-glass study*. Therefore, we believe that the impact of our work is not limited in the field of spintronics, but could be expanded to other research fields, highlighting the broad interest of our work.

We have added above discussion in the revised manuscript.

I also have specific several comments on the manuscript that I feel would improve the article.

1. The results of figure 2 are of hysteresis (quasi-static) loops and those of figure 3 non-equilibrium. The responses of the moments from the different bands will be different depending on the time-scale, so I am not convinced by the comparability between the two sets of results.

→ We thank the reviewer for his/her fruitful comment. As the reviewer mentioned, Fig. 3 shows the TR-MOKE results in non-equilibrium regime, which differs from the quasi-static Hall and VSM measurement in Fig. 2. However, we think that TR-MOKE can capture the variation of magnetic

moment occurring in quasi-static regime because of the following reason. Figure R1 shows schematic illustrations of magnetization after pump laser incidence. TR-MOKE measures the laser-driven ultrafast demagnetization process. Therefore, the measured variation of Kerr angle represents the amount of magnetic moment reduction, $\Delta m = \eta m$, where m is the magnetic moment before pump pulse incidence and η is the demagnetization efficiency. During the TR-MOKE measurement, experimental temperature and pump fluence remained constant within an error of less than 1%. Hence, the time evolution of TR-MOKE at a fixed delay time should represent the slow variation of magnetic moment (e.g, change of magnetization direction from φ_1 to φ_2 in Fig. R1). Therefore the TR-MOKE results can capture the quasi-static response of magnetic moment.

Fig. R1. Schematic illustration of magnetic moment configurations during the TR-MOKE measurement. The slow change of magnetization angle from φ_1 to φ_2 can be reflected in the TR-MOKE data.

Nevertheless, we agree with the reviewer that it should be verified to the comparability between Fig. 2 and Fig. 3 for the same dynamic regime. To this end, we performed the additional experiment as follows.

There are two main points that we would like to claim through the TR-MOKE in Fig. 3: First, the magnetic moments at a deeper energy level are more susceptible to in-plane magnetic field (Figs. 3(b)-3(d)). Second, the magnetic moment of TbCo shows spin-glass-like slow dynamics (Figs. 3(e)-3(f)). To confirm that this is also in case for quasi-static regime, we performed *static-MOKE* measurement. Figure R2 shows the variation of in-plane static-Kerr rotation while sweeping in-plane magnetic field for $\lambda = 400$ nm (black) and $\lambda = 800$ nm (red) (sweep rates were the same for both measurements). The

measurement setup is identical to that in Fig. 3a of the manuscript. The in-plane static-Kerr rotation was obtained by extracting the odd component of Kerr signal, as we explained in the manuscript. As can be seen, the variation of Kerr angle for the in-plane magnetic field is approximately 8 times larger for $\lambda = 400$ nm than that for $\lambda = 800$ nm. This means that the magnetic moment at deeper energy level is more susceptible to the in-plane magnetic field *even at quasi-static regime*, which is consistent with the TR-MOKE results in Fig. 3 of manuscript. We note that the larger noise for $\lambda = 400$ nm originates from the wavelength-dependent optical sensitivity of our detector, as shown in Fig. R3 (In this study, we used DET36A which is pre-improved version of DET36A2).

Fig. R2. In-plane Kerr rotation angle as a function of external in-plane magnetic field for $\lambda = 400$ nm (black) and $\lambda = 800$ nm (red), measured by static-MOKE.

Fig. R3. Wavelength dependent sensitivity of optical detector.

We also carried out a static-MOKE experiment to check the spin-glass-like slow dynamics, which occurs in the time scale of mins to hours. However, we found that it was very hard to capture the long-time variation of static-Kerr rotation because of the long-time drift and large noise in static-MOKE setup (we note that such a long-time drift is absent in the TR-MOKE because the TR-MOKE probes the relative Kerr angle change after pump pulse incidence). To overcome this issue, we have performed a different quasi-static measurement, i.e., direct measurement of magnetic moment variation, by using MPMS (magnetic properties measurement system).

Figure R4 shows the MPMS results obtained from TbCo film. In Fig R4(a), we first applied magnetic field of $H = +1$ T along the in-plane direction for 30 minutes and subsequently reduced the magnetic field to $H = +1$ mT. After that, we measured the magnetic moment in time. $|M(0)|$ is the absolute value of magnetization of the TbCo film immediately after the magnetic field being reduced to +1 mT, and $M(t)$ corresponds to the magnetization variation in time t thereafter. As can be seen, the total magnetic moment decreases slowly over a long time, which indicates the slow relaxation of magnetic moment. The same trend is observed in experiment with opposite magnetic field (Fig. R4(b)), confirming that the slow relaxation is not caused by possible drift in the setup. These *quasi-static* experiments confirm the spin-glass-like slow spin dynamics that we have observed with TR-MOKE measurement. Thus, these newly performed experiments provide the direct proof that the referee requested and thereby substantially enhance and make our manuscript much more compelling.

We have added above discussion with additional data in our revised Supplementary Note 7.

Fig. R4. Magnetic moment as a function of time measured by MPMS for (a) positive field and (b) negative field.

2. I found that referring to the work as “energy dependent” a little (I think unintentionally) misleading. From the title I had the impression that the measurements would be probing something across a spectrum of energies, rather than just a select few.

→ We thank the reviewer for this comment. Following the reviewer’s comment, we revised the title as “observation of spin-glass-like characteristics in ferrimagnetic TbCo through energy-level-selective approach”. We believe that, thanks to the reviewer’s comment, the new title more clearly expresses our key findings.

3. I could not follow what exactly was done in figure 2e. The figure is unclear and the description in the text I found didn’t really help. There is talk of switching not being instantaneous, which, again, I could not follow. Why would one expect a switching process through the application of a field to be instantaneous? Why did it take 10 minutes? What is the process by which this occurred? More detail needs to be provided here.

→ We thank the reviewer for this careful comment. We think that the reviewer meant Fig. 3(e), not Fig. 2(e). Through Fig. 3(e), we tried to explain our key finding that *the reaction of magnetic moment to the magnetic field is extremely slow in ferrimagnetic TbCo*.

We first explain the measurement details of Fig. 3(e). The black data in Fig. 3(e) represents the laser-induced ultrafast demagnetization and recovery profile, which was measured by 400 nm pump/400 nm probe TR-MOKE. Here, we applied in-plane magnetic field of +600 mT and waited for 1.5 hours before the measurement. The reason for the long waiting time is because the saturation of magnetic moment takes a long time, as we will discuss later. After measuring the black data, we reversed the magnetic field to the opposite direction, i.e., $H = -600$ mT, and then measured the Kerr angle at $t = 4$ min, 11 min, 18 min, 25 min after applying the magnetic field. This is because the Kerr angle does not reverse instantaneously, but takes long time to be fully saturated. Here, we only monitored the Kerr angle at fixed delay time of 55 ps (blue symbols), because the full profile measurement takes time and the Kerr angle gradually changing even during the measurement. After $t = 40$ min, we confirmed that the Kerr angle does not change anymore, so we measured the full Kerr angle profile (red data). Figure 3(f) shows the gradual change of Kerr angle in time (which corresponds the symbols in Fig. 3(e)), highlighting that the Kerr angle is gradually reversed in a time scale of a few tens of minutes.

To visualize the slow variation of Kerr angle more directly, we measured a real time variation of Kerr angle after reversing the magnetic field. Figure R5 (a) shows the real time evolution of TR-MOKE results at fixed delay time of 55 ps. The noise level is high because the average process is not possible

in real time measurement. It is clear that the Kerr angle gradually changes in a time scale of a few tens of minutes, which is consistent with Fig. 3(f) of manuscript. We note that this data does not come from the drift. (See Figure R5 (b) after saturation)

Fig. R5. Real time TR-MOKE data at fixed delay time of 55ps (a) during time saturation and (b) after time saturation.

The slow variation of Kerr angle indicates that the magnetic moments of TbCo respond to the magnetic field very slowly. We further discuss the time scale for the response of magnetic moment. When an in-plane magnetic field is applied to a ferromagnet with perpendicular magnetic anisotropy, one can expect that the magnetic moment tilts along the in-plane field direction. In this process, the time scale is typically determined by the strength of magnetic anisotropy. In usual ferromagnets, the anisotropy energy is around GHz range, so that the response of magnetic moment to the magnetic field is expected to occur in a nanosecond time scale. Therefore, the Kerr angle reversal after reversing the magnetic field should occur almost instantaneously. However, what we observed in TbCo is completely different. The reversal of Kerr angle takes extremely long time as long as a few tens of minutes (we note that we have also confirmed this slow relaxation of moment by using quasi-static MPMS measurement as shown in Fig. R4). This slow relaxation of magnetic moment in ferrimagnet is a surprising result that has never been expected.

The slow dynamics is a typical feature of spin-glass systems, such as single molecular magnets (SMMs) or magnetic nanoparticles (NPs). As shown in Fig. 3d of [M. Mannini *et al.*, *Nat. Mater.* **8**, 194 (2009)], where the time dependent dichroic signal of SMM was measured after reversing magnetic field, the response of magnetic moment to the magnetic field is extremely slow in spin-glass systems. Therefore, what we observed in TbCo may be related to the spin-glass characteristics. The spin-glass is a system where multiple ground states exist, as shown in Fig. R6. Therefore, the thermal hopping

between the ground states, which typically takes a long time, can occur. This should be the origin of slow dynamics. Furthermore, such spin-glass characteristic can be verified by lowering temperature, because the thermal hopping is strongly suppressed at low temperature and thus, the moment freezing can be observed. This is indeed what we observed in Fig. 4 of our manuscript using THz emission. Therefore, we concluded that the ferrimagnetic TbCo exhibits the spin-glass-like properties.

Fig. R6. Schematics of multiple ground states in spin-glass system.

We further discuss the possible physical origin of the spin-glass-like properties of TbCo. Since there is no universal Hamiltonian to describe spin-glass [Binder, K. *et al.*, *Rev. Mod. Phys.* **58**, 801–976 (1986).], the origin of spin-glass has so far been studied individually according to the specific system. For example, H. Maletta studied the spin glass characters in $\text{Eu}_x\text{Sr}_{1-x}\text{S}$ based on dipolar interaction [*J. Appl. Phys.* **53**, 2185 (1982)] and P. M. Levy, *et al.* studied CuMn based on DM (Dzyaloshinskii–Moriya) type interaction [*Phys. Rev. B* **23**, 4667–4690 (1981)]. The fundamental reason for the spin-glass-like properties of our TbCo is not perfectly understood at the present stage and requires further studies. However, we think that it is possibly due to the random anisotropy of Tb. According to [E. M. Chudnovsky *et al.*, *Phys. Rev. B* **33**, 251 (1986)], the correlated spin glass state can be achieved under the condition for $H_{\text{applied}} < \frac{H_r^4}{H_{\text{ex}}^3}$, where H_{applied} is the applied field, H_r is the random anisotropy field and H_{ex} is the exchange field. Therefore, the spin-glass properties can appear when the H_r is bigger than $(H_{\text{applied}}H_{\text{ex}}^3)^{1/4}$. With the literature value of $H_{\text{ex}} = 138 \text{ T}$ [S. Kervan *et al.*, *J. Supercond. Novel Magn.* **24**, 819 (2011)] and the experimental value of $H_{\text{applied}} = 600 \text{ mT}$, we can roughly estimate the correlated spin-glass condition as $H_r > 35.4 \text{ T}$ (Note that the value of H_{ex} has some

variation depending on the literatures [P. Hansen *et al.*, *J. Appl. Phys.* **66**, 756 (1989), S. Kervan *et al.*, *J. Supercond. Novel Magn.* **24**, 819 (2011)]. Although the 35.4T is far beyond the accessible magnetic field in the laboratory, we have checked if the H_r could really be that large.

To check the H_r of Tb, we prepared the Tb-only sample by using same DC magnetron sputtering procedure [exact structure is Si/SiO₂/Tb(250 nm)/Ta(1.5 nm); here Ta capping was used for preventing oxidation], and measured the field-dependent magnetization variation. Figure R7 shows raw data of M-H curve for Tb-only sample at $T = 5$ K. As can be seen, the magnetic moment does not saturate even for $H = 7$ T which is the maximum magnetic field in the measurement setup (Magnetic Properties Measurement System, MPMS). We note that the positive slope does not originate from the background of substrate, because the diamagnetic Si/SiO₂ substrates produces the negative slope of background signal. The non-saturating positive slope in M-H curve indicates that the anisotropy of randomly deposited Tb is much greater than 7 T. We note that the literature value of Tb anisotropy is much bigger than this, as stated in p. 220 of [Wohlfarth, Erich Peter, ed. Handbook of magnetic materials. Vol. 2. Elsevier, (1986)] that “*the field required to magnetize Tb in the c direction is of the order of a million to ten million oersteds...*”. Considering that our Tb has sputter-deposited polycrystalline structure, it is reasonable to think that the Tb has a strong random anisotropy field of the magnitude of the order of tens of Tesla. Therefore, the result in Fig. R7 together with the literature suggests that the Tb indeed shows a strong random anisotropy, which could meet the spin-glass condition.

We have added above discussion in our revised Supplementary Note 11.

Fig. R7. M-H curve of Tb-only sample at $T = 5$ K.

4. Also related to point 3. The discussion around the spin-glass characteristics was also not clear.

There is no real introduction to the reader about spin-glasses, which is rather niche. The presence of the spin-glasses is introduced to explain the slow “switching” behaviour, which, as mentioned in point 3 is not clear.

→ We thank the reviewer for the valuable suggestion. The detailed introduction of spin glass, which we have discussed in the previous answer, has been inserted in the revised manuscript as follows.

p. 8 L. 14 *“The slow spin dynamics is a typical feature of spin-glass systems³². Therefore, what we observed in TbCo may be related to the spin-glass characteristics. The spin-glass is a system where multiple ground states exist. In this case, the thermal hopping between the ground states, which typically takes a long time, can occur. This seems to be the origin of slow dynamics. Furthermore, such spin-glass characteristic can be verified by lowering temperature, because the thermal hopping is strongly suppressed at low temperature and thus, the moment freezing can be observed.”*

Minor Points

1. The authors refer to TbCo as being antiferromagnetic, e.g. line 32/33, “As a result, the antiferromagnetic spin configuration”. It does not have an antiferromagnetic spin configuration, but a ferrimagnetic with two sublattices with antiferromagnetic exchange coupling. This point should be made clearer throughout.

→ We thank the reviewer for kind suggestion. Following the reviewer’s suggestion, we revised the “antiferromagnetic spin configuration” into the “two sublattices with antiferromagnetic exchange coupling”.

2. The reference to a “submoment” is not clear.

→ We agree with the reviewer’s concerns. The word “submoment” has been changed to “moment from each sub-lattice”.

3. In figure 3 or its caption, mention which material is being studied.

→ We thank the reviewer for kind suggestion. TbCo film was used in Fig. 3. We revised the manuscript to clearly represent the material.

Reviewer #2 (Remarks to the Author):

Ji-Ho Park et al investigated the magnetic moment configuration of ferrimagnetic TbFe alloys using various methods including Hall effect, VSM, TR-MOKE, and THz emission, and distinguished the sub-moments of Co3d and Tb4f under pump/probe light with wavelength of 800 and 400 nm, respectively. This method seems to be an effective way, but the following issues should be addressed first before publication.

1. For TR-MOKE, the authors say that the IP Kerr signal at 400 nm is more dominant than the OPP one, indicating the magnetic moments at a deeper energy level are more susceptible to HIP. What is the relation of the IP/OPP Kerr signal with the magnetic moments at fermi/deeper energy level?

→ We thank the reviewer for this valuable comment. We explain the relation between the Kerr angle and magnetic moment in a more quantitative way. Figure R8 shows the schematic illustration of magnetic moment configurations of Co and Tb in ferrimagnetic TbCo under in-plane magnetic field. Here, Co and Tb directed in opposite direction along out-of-plane (OOP) direction due to antiparallel exchange interaction, while they are both tilted along the same in-plane (IP) direction due to presence of in-plane magnetic field (the possibility of spin canting will be discussed in the response to the reviewer #3). Since the TR-MOKE measures the laser-driven ultrafast demagnetization process, the measured variation of Kerr angle represents the amount of magnetic moment reduction, $\Delta m = \eta m$, where m is the magnetic moment and η is the demagnetization efficiency. We assume that the variation of Kerr angle, $\Delta\theta$, is proportional to that of magnetic moment Δm , as it is generally accepted in the community. We note that any other parameters that can affect the Kerr angle [for other parameters, see Eqs. (18) and (19) of You *et al.*, *Appl. Phys. Lett.* **69**, 1315 (1996)] cannot explain the large difference in $\Delta\theta_K^{IP}/\Delta\theta_K^{OOP}$ for different wavelengths shown in Fig. 3(d). Therefore, the $\Delta\theta_K^{IP}/\Delta\theta_K^{OOP}$ is mainly determined by the variation of magnetic moment.

We first check the relation of $\Delta\theta_K^{IP}/\Delta\theta_K^{OOP}$ with the magnetic moments near the Fermi energy level. In TbCo, the magnetic moment near the Fermi level is largely dominated by the Co 3d moment. Therefore, the Kerr angle ratio $\Delta\theta_K^{IP}/\Delta\theta_K^{OOP}$ can be expressed by

$$\frac{\Delta\theta_K^{IP}}{\Delta\theta_K^{OOP}} \sim \frac{\Delta m_{IP}^{Co}}{\Delta m_{OOP}^{Co}} = \frac{\Delta m^{Co} \sin\varphi_{Co}}{\Delta m^{Co} \cos\varphi_{Co}} = \tan\varphi_{Co}, \quad (R1)$$

where φ_{Co} is the tilting angle of Co moment by the in-plane magnetic field. This indicates that the Kerr angle ratio between IP and OOP direction near Fermi level is mainly governed by the tilting angle of Co, which approximately explains the TR-MOKE results for $\lambda = 800$ nm (here, ‘approximately’

means that the laser with $\lambda = 800$ nm can probe not only the Fermi level but also the deeper energy level down to $E_{\lambda=800nm} = 1.6$ eV).

Unlike the magnetic moment near Fermi level, the magnetic moments at deeper energy level is much more affected by the Tb moment, because the $4f$ level of Tb lies at deeper energy level. Therefore, if we use a laser of $\lambda = 400$ nm ($E_{\lambda=400nm} = 3.2$ eV), it can probe not only the Co moment near Fermi level but also the Tb moment at deeper energy level. Then, the Kerr angle ratio $\Delta\theta_K^{IP}/\Delta\theta_K^{OOP}$ can be expressed by

$$\frac{\Delta\theta_K^{IP}}{\Delta\theta_K^{OOP}} \sim \frac{\Delta m_{IP}^{Co} + \Delta m_{IP}^{Tb}}{\Delta m_{OOP}^{Co} - \Delta m_{OOP}^{Tb}} = \frac{\Delta m^{Co} \sin\varphi_{Co} + \Delta m^{Tb} \sin\varphi_{Tb}}{\Delta m^{Co} \cos\varphi_{Co} - \Delta m^{Tb} \cos\varphi_{Tb}}. \quad (R2)$$

Here, φ_{Tb} is the tilting angle of Tb moment. We note that the plus (minus) sign at numerator (denominator) indicates that the magnetic moments of Co and Tb align along the same (opposite) direction for IP (OOP) direction. This relation explains the TR-MOKE results for $\lambda = 400$ nm.

To be more quantitative, we roughly estimate the in-plane field-driven tilting angle of each magnetic moment using above equations. According to Fig. 3(d), the $\Delta\theta_K^{IP}/\Delta\theta_K^{OOP} \sim 0.1$ for $\lambda = 800$ nm (orange symbols in Fig. 3(d)). Therefore, the tilting angle of Co moment is approximately $\varphi_{Co} = 6^\circ$ based on Eq. (R1). On the other hand, $\Delta\theta_K^{IP}/\Delta\theta_K^{OOP} \sim 10$ for $\lambda = 400$ nm (blue symbols in Fig. 3(d)). If we assume that $\Delta m^{Co} \sim \Delta m^{Tb}$ because the composition of our sample ($Co_{75}Tb_{25}$) is near the compensation point, then the tilting angle of Tb is approximately $\varphi_{Tb} = 17^\circ$ based on Eq. (R2). Therefore, the different Kerr angle ratio of $\Delta\theta_K^{IP}/\Delta\theta_K^{OOP}$ for different wavelengths manifests that the Tb moment, which lies at deeper energy level, is more susceptible to the in-plane magnetic field. Therefore, we believe that the Kerr effect measurement in Fig. 3 corroborates the Hall and VSM measurement in Fig. 2.

We have added above discussion in our revised Supplementary Note 4.

Fig. R8. Schematic illustration of magnetic moment configurations of Co and Tb in ferrimagnetic TbCo under

in-plane magnetic field.

2. In caption of Figure 3, with an in-plane magnetic field $H_{IP} = 500$ mT, confirming the saturation of the magnetic moment. Please see the VSM curve in Fig.2(i), it shows not saturated even at $H_{IP} = 600$ mT. In fact on the contrary, if saturated, the magnetic moment is totally in the film plane, then the OPP Kerr signal should not be detectable.

→ We are very sorry for causing the misunderstanding. The “saturation” in the caption of Fig. 3 does not mean the “saturation of magnetic moment along the in-plane direction”, but means the “stabilization of the magnetic moment after a sufficiently long time”. As the TbCo exhibits the spin-glass-like slow dynamics, it takes a long time (min to hours) to move to local equilibrium state. When we performed the experiment in Fig. 3, we waited for about 1.5 hours after applying magnetic field for the magnetic moment to be stabilized. At this stabilized state, the Tb and Co moment tilts from the out-of-plane, but does not fully align along the in-plane direction, because the in-plane magnetic field is not sufficiently strong to overcome the anisotropy of Tb and Co, as the reviewer mentioned. Therefore, the out-of-plane component of Kerr rotation can be detectable.

We have revised the caption as “... *and confirming that the Kerr angle is stabilized.*”

3. The whole article should make clear the discription of Kerr rotation/angle with the Kerr signal. They are two different parameters.

→ We thank the reviewer for this careful comment. We have revised the arbitrary unit Kerr signal into Kerr angle throughout the manuscript.

4. About the figures of 3(b) and 3(c), the changes of Kerr angle at the beginning are not clear. They need enlarged insets with smaller delay time within several picoseconds.

→ We thank the reviewer for this comment. Figure R9 shows the early stage of ultrafast demagnetization process observed in TbCo.

We note that in our manuscript we did not focus on this early stage of ultrafast demagnetization, but rather focused on the relaxation regime ($t > 40$ ps). This is because the early stage of ultrafast demagnetization comprises the complex behavior of magnetic moments [e.g., “two step demagnetization” occurs at $t < 15$ ps, A. R. Khorsand *et al.*, *Phys. Rev. Lett.* **110**, 107205 (2013)]. In addition, the early stage of ultrafast demagnetization is not relevant to our main findings. In this

manuscript, we report that the ferrimagnetic TbCo exhibits the spin-glass-like slow dynamics, *which occurs at quasi-static time scale*. Therefore, we focused on the longer time scale rather than the ultrafast demagnetization regime. We note that we also provided the *quasi static-measurement results* during the revision process (see the response to the reviewer #1 and Figs. R2 and R4).

Although we did not focus on the ultrafast demagnetization of TbCo, it is worth to investigate it in future studies, because the origin of ultrafast demagnetization is still highly controversial issue. After the first discovery of ultrafast demagnetization at Ni in 1995 [E. Beaurepaire *et al.*, *Phys. Rev. Lett.* **76**, 4250 (1996)], various arguments have been put forward to explain the ultrafast demagnetization, such as coherent interaction with the photon field [J.-Y. Bigot *et al.*, *Nat. Phys.* **5**, 515 (2009)], spin super-diffusion [P. Tengdin *et al.*, *Sci. Adv.* **4**, eaap9744 (2018)], ultrafast magnon scattering [E. Carpene *et al.*, *Phys. Rev. B* **78**, 174422 (2008)] and electron impurity scattering [B. Koopmans *et al.*, *Phys. Rev. Lett.* **95**, 267207 (2005)]. The origin of ultrafast demagnetization is even more complicated in rare earth metals because effect of lattice on the demagnetization is different depending on the rare earth element [B. Frietch *et al.*, *Sci. Adv.* **6**, eabb1601 (2020)]. Therefore, our data of TbCo may provide an experimental clue for better understanding of ultrafast demagnetization process of ferrimagnetic systems, which we believe beyond the scope of our paper.

Fig. R9. Early stage of ultrafast demagnetization for 800nm and 400nm pump and probe laser at 305K. Green dots are in plane signal and Black dots are out of plane signal.

Reviewer #3 (Remarks to the Author):

In this manuscript, the authors have studied and compared the magnetic properties of ferrimagnetic TbCo alloy with ferromagnetic Co films by the AHE, VSM, TR-MOKE, and THz emission measurements. They found the variation of the Hall signal for TbCo is extremely small, only 4% of the total Hall signal change from perpendicular (HOOP) to in-plane (HIP) field of 600 mT. In contrast, the VSM loops indicate that the total magnetization variation caused by HIP is comparable to that caused by HOOP. Based on the difference between the in-plane Hall and VSM results, the authors conclude that the magnetic moments respond differently to the external magnetic depending on the energy level of 3d and 4f, and the Tb submoment at a deeper energy level is more easily changed by the magnetic field than the Co submoment near the EF.

The reviewer has some concerns about these opinions. Firstly, we know the AF coupling in TbCo is very strong, so the Co magnetic moments should rotate simultaneously with the magnetic moments of Tb. It is not possible to keep Co undisturbed when Tb is rotated by applying HIP in the range of just \square 600 mT.

→ We thank the reviewer for this valuable comment. As the reviewer mentioned, Co and Tb are bound by antiparallel exchange coupling, so one can expect that the Co magnetic moments should rotate simultaneously with the magnetic moments of Tb. This is based on the *rigid two-arrow model* where the magnetic moments of Tb and Co are treated as antiferromagnetically coupled rigid arrows. However, what we found through this study is that *this simple model cannot explain our experimental results*.

We first discuss the magnetic energies of ferrimagnetic TbCo. The magnetic structure of TbCo can be understood by comparing the magnitudes of the terms in the Hamiltonian [J.M.D. Coey *et al.*, *Phys. Rev. Lett.* **36**, 1061 (1976)],

$$-H = \sum_i V_{1i} + \sum_j V_{2j} + \sum_{i>i'} g_{11} J_i \cdot J_{i'} + \sum_{j>j'} g_{22} J_j \cdot J_{j'} + \sum_{i,j} g_{12} J_i \cdot J_j \quad (R3),$$

where the first two terms represent the anisotropy energies of Tb and Co and the third and fourth terms the ferromagnetic exchange energies of Tb and Co, respectively. Last term denotes the antiferromagnetic exchange energy between Tb and Co. i and j run over the Tb (1) and Co (2) sublattices, respectively. Therefore, the spin configurations of Tb and Co moments are determined by the relative energies described in Eq. (R3). According to the literatures, the value of exchange energy has a variation to some extent. However, although there is a variation in the value of exchange energy, the antiferromagnetic exchange energy between Tb and Co is usually one order smaller than the exchange energy of Co atoms [Table 2 in S. Kervan *et al.*, *J. Supercond. Nov. Magn.* **24**, 819 (2011), Table II in M. Mansuripur *et al.*, *IEEE Trans. Magn.* **MAG-22**, 33 (1986)], which allows the small

canting of magnetic moment by magnetic field. Furthermore, the strong random anisotropy of Tb makes the Tb moment direction dispersed [see Fig. R7 and discussion therein for the strength of random anisotropy of Tb]. Therefore, we think that there could be a canting of magnetic moments between Tb and Co.

Secondly, the observed similar magnetization changes of IP and OOP VSM loops for the TbCo sample are not reasonable, since the HOOP loop has been already saturated while the HIP loop seems far from saturation in Fig.2. The unsaturated IP loop suggests that the magnetization variation should be much smaller than the OOP loop. I am afraid that the small net magnetization of TbCo is hardly tilted by HIP and the obtained similar value in Fig. 2 may probably arise from the background signal which is not correctly subtracted from the raw data.

→ We thank the reviewer for this comment. Based on the *rigid two-arrow model*, where the magnetic moments of Tb and Co are treated as antiferromagnetically-coupled rigid arrows, magnetization change for IP cannot exceed the saturated magnetization for OOP, as the reviewer mentioned. However, as we explained in the previous answer, *this simple model is not sufficient to explain the magnetic moment configuration in TbCo system*. This is because the TbCo exhibits the sperimagnetic nature [Hussain, R., et al. *Journal of Superconductivity and Novel Magnetism* 32.12, 4027-4031 (2019), Figs. 8 and 9 of N.H. Duc *et al.*, “chapter 2 Magnetoelasticity in Nanoscale Heterogeneous Magnetic Materials” *Handbook of Magnetic Materials*, **14**, 89 (2002) and J. Yu *et al.*, *J. Mag. Mag. Mater.* **487**, 165316 (2019)] and spin canting due to weak antiferromagnetic exchange interaction between Tb and Co. In the following, we explain how these properties can explain the VSM results.

We first discuss the background effect that the reviewer concerns. In our measurement, the background subtraction was conducted very carefully, because it can affect the analysis, as the reviewer mentioned. Generally, the linear background in VSM data originates from the diamagnetic response of the substrate. Therefore, one can simply remove it by subtracting the linear component from the measured VSM curve. In our measurement, however, this subtraction process is inappropriate because the non-saturating magnetic moment could give the linear signal in VSM data. To overcome this difficulty, we have devised a more accurate background subtraction process. We prepared two samples: one is the bare substrate and the other is the TbCo/substrate. Here, the substrates for two samples are almost identical with each other (thickness, width and length differences between two substrates were less than 1%). We then performed the VSM measurement for two samples using the same experimental protocol. Finally, we subtracted the signal of bare substrate from that of TbCo/substrate. Figures R10(a) and R10(b) show the raw VSM signals for OOP (a) and IP (b) for TbCo/substrate (black) and bare substrate (red). For OOP signal (R10(a)), the linear component from the TbCo/substrate and bare substrate has same slope, resulting in an almost perfect background subtraction, as shown in Fig. R10(c).

However, for IP signal (R10(b)), the slopes for two samples are different and thus, the linear component cannot be completely removed, as shown in Fig. R10(d). This means that the remained linear slope in Fig. R10(d) comes from the TbCo itself.

Figure R10. Magnetization as a function of (a) OOP (b) IP magnetic field measured by VSM. Black (red) data shows the result for TbCo/substrate (substrate-only). (c), (d) M-H curve obtained by subtracting the red data from the black data in Figs. R10(a) and R10(b).

Next, we will discuss how the unsaturated IP moment can be greater than the saturated OOP moment. What we found from the Hall and VSM measurement (Fig. 2) as well as the MOKE measurement (Fig. 3) was that the Tb moment is more susceptible to the in-plane magnetic field than the Co moment. Figure R11 shows the schematic illustration of magnetic moment configurations of Co and Tb in ferrimagnetic TbCo under in-plane magnetic field. We note that the Tb moment directions are spread out due to the sperimagnetic nature of TbCo [see Figs. 8 and 9 of N.H. Duc *et al.*, “chapter 2 Magnetoelasticity in Nanoscale Heterogeneous Magnetic Materials” *Handbook of Magnetic Materials*, **14**, 89 (2002)]. Here, Co and Tb directed in opposite direction along out-of-plane (OOP) direction due to antiferromagnetic exchange interaction, while they are both tilted along the same in-plane (IP) direction due to presence of in-plane magnetic field. We note this magnetic moment configuration has recently been evidenced in other report [Figs. 3(c) and 3(f) of J. Yu *et al.*, *J. Mag. Mag. Mater.* **487**, 165316 (2019)]. Then the out-of-plane (OOP) magnetic moment under the OOP field, M_{OOP} , is $M_{OOP} = m_{Co} - m_{Tb}$, because two moments directed to opposite direction. However, the total in-plane

(IP) magnetic moment under IP field, M_{IP} , is $M_{IP} = m_{Co} \sin \varphi_{Co} + m_{Tb} \sin \varphi_{Tb}$. Here, importantly, two moments, m_{Co} and m_{Tb} , contribute to the total moment in a subtractive way for OOP while in an additive way for IP. Therefore, the saturated OOP moment can be small near the ferrimagnetic compensation point (which is in our case), while the unsaturated IP moment can be greater than the saturated OOP moment and can even be increased.

Fig. R11. Schematic of magnetic moment configuration in TbCo without (left) and with (right) in-plane magnetic field. The bottom arrows denote the average value of Co and Tb moments.

Lastly, for the control sample of Co film, the Hall and VSM results are very consistent, both can be switched by applying an IP field of 600mT, suggesting that the Co moments are actually not difficult to be changed. According to my knowledge, whether the magnetization orientation can be changed or not, relies on the total net magnetization, the effective magnetic anisotropy field as well as the applied field.

→ We thank the reviewer for this fruitful comment. We agree with the reviewer that the magnetization orientation depends on the total net magnetization, effective magnetic anisotropy, as well as applied field. However, we think such idea can be applied to a simple system, where the magnetic moment can be expressed by macroscopic rigid arrows such as ferromagnetic Co. As we explained in the previous answers, we found in this study that the simple model is insufficient to explain the magnetic moments of ferrimagnetic TbCo, because the Tb moment direction can be spread out and the Tb and Co moments

can be canted with each other by external magnetic field due to the weak antiferromagnetic exchange interaction. This implies that the Co moment behaves differently in ferromagnetic Co and in ferrimagnetic TbCo.

To be more quantitative, we roughly estimate how the magnetic moments of TbCo respond to the in-plane magnetic field. To this end, we use the TR-MOKE result in Fig. 3(d) of manuscript. Figure R12 shows the schematic illustration of magnetic moment configurations of Co and Tb in ferrimagnetic TbCo under in-plane magnetic field, as we discussed in the previous answer. Since the TR-MOKE measures the laser-driven ultrafast demagnetization process, the measured variation of Kerr angle represents the reduction of magnetic moment, $\Delta m = \eta m$, where m is the magnetic moment and η is the demagnetization efficiency. We assume that the variation of Kerr angle, $\Delta\theta$, is proportional to that of magnetic moment Δm , as it is generally accepted in the community.

We first check the relation of $\Delta\theta_K^{IP}/\Delta\theta_K^{OOP}$ with the magnetic moments near the Fermi energy level. In TbCo, the magnetic moment near the Fermi level is largely dominated by the Co $3d$ moment. Therefore, the Kerr angle ratio $\Delta\theta_K^{IP}/\Delta\theta_K^{OOP}$ can be expressed by

$$\frac{\Delta\theta_K^{IP}}{\Delta\theta_K^{OOP}} \sim \frac{\Delta m_{IP}^{Co}}{\Delta m_{OOP}^{Co}} = \frac{\Delta m^{Co} \sin\varphi_{Co}}{\Delta m^{Co} \cos\varphi_{Co}} = \tan\varphi_{Co}, \quad (R4)$$

where φ_{Co} is the tilting angle of Co moment by the in-plane magnetic field. This indicates that the Kerr angle ratio between IP and OOP direction near Fermi level is mainly governed by the tilting angle of Co, which approximately explains the TR-MOKE results for $\lambda = 800$ nm. (here, ‘approximately’ means that the laser with $\lambda = 800$ nm can probe not only the Fermi level but also the deeper energy level down to $E_{\lambda=800nm} = 1.6$ eV).

Unlike the magnetic moment near Fermi level, the magnetic moments at deeper energy level is more affected by the Tb moment, because the $4f$ level of Tb lies at deeper energy level. Therefore, if we use a laser of $\lambda = 400$ nm ($E_{\lambda=400nm} = 3.2$ eV), it can probe not only the Co moment near Fermi level but also the Tb moment at deeper energy level. Then, the Kerr angle ratio $\Delta\theta_K^{IP}/\Delta\theta_K^{OOP}$ can be expressed by

$$\frac{\Delta\theta_K^{IP}}{\Delta\theta_K^{OOP}} \sim \frac{\Delta m_{IP}^{Co} + \Delta m_{IP}^{Tb}}{\Delta m_{OOP}^{Co} - \Delta m_{OOP}^{Tb}} = \frac{\Delta m^{Co} \sin\varphi_{Co} + \Delta m^{Tb} \sin\varphi_{Tb}}{\Delta m^{Co} \cos\varphi_{Co} - \Delta m^{Tb} \cos\varphi_{Tb}}. \quad (R5)$$

Here, φ_{Tb} is the tilting angle of Tb moment. We note that the plus (minus) sign at numerator (denominator) indicates that the magnetic moments of Co and Tb align along the same (opposite) direction for IP (OOP) direction. This relation explains the TR-MOKE results for $\lambda = 400$ nm.

Using Eqs. R(4) and R(5) together with the experimental data in Fig. 3(d), we can roughly estimate the in-plane field-driven tilting angle of each magnetic moment. According to Fig. 3(d), the

$\Delta\theta_K^{IP}/\Delta\theta_K^{OOP} \sim 0.1$ for $\lambda = 800$ nm (orange symbols in Fig. 3(d)). Therefore, the tilting angle of Co moment is approximately $\varphi_{Co} = 6^\circ$ based on Eq. (R4). On the other hand, $\Delta\theta_K^{IP}/\Delta\theta_K^{OOP} \sim 10$ for $\lambda = 400$ nm (blue symbols in Fig. 3(d)). If we consider that $\Delta m^{Co} \sim \Delta m^{Tb}$ because the composition of our sample ($Co_{75}Tb_{25}$) is near the compensation point, then the tilting angle of Tb is approximately $\varphi_{Tb} = 17^\circ$ based on Eq. (R5). This result suggests that the magnetic moment of Co in TbCo is not fully saturated along the in-plane direction by $H_{ip} = 500$ mT, which is different from the Co moment in ferromagnetic Co. Furthermore, the above discussion implies that the Tb moment is more susceptible to the magnetic field, and thus the Co and Tb moment can be canted with each other in TbCo system. Therefore, the magnetic moments configuration as well as their dynamics should be more carefully treated in ferrimagnets, which we believe is the reason why our research should urgently be known to the community.

Fig. R12. Schematic illustration of magnetic moment configurations of Co and Tb in ferrimagnetic TbCo under in-plane magnetic field.

Reviewers' Comments:

Reviewer #1:

Remarks to the Author:

This reviewer would like to thank the authors for their comprehensive response to my, and the other reviewers' comments. This has greatly improved my understanding and interpretation of the work and I believe the article will be much improved for the general reader. I believe the work will stimulate discussion and further work in the magnetism community. From my side, the authors have addressed all of my points, however, I have one comment, which is on the use of the term rigid arrow to describe previous works, specifically, I find a little unclear. In the work by Khorsand they are sensitive to measurements of the change in the length of the magnetisation, which I would argue is not rigid.

Reviewer #2:

Remarks to the Author:

I prefer using "Kerr signal" in figures because Kerr angle is a special angle which should be measured by rotating the polarizer till the maximum signal appears, i.e. the Kerr angle is the rotation of polarization of reflected light. Normally we measure the Kerr signal by setting a polarizer at a suitable angle, just detecting the variation of reflected light intensity at a certain angle of polarizer. Then, I recommend this article published in nc.

Reviewer #3:

Remarks to the Author:

I would like to recommend the revised version for publication in this journal

Reviewer #1 (Remarks to the Author):

This reviewer would like to thank the authors for their comprehensive response to my, and the other reviewers' comments. This has greatly improved my understanding and interpretation of the work and I believe the article will be much improved for the general reader. I believe the work will stimulate discussion and further work in the magnetism community. From my side, the authors have addressed all of my points, however, I have one comment, which is on the use of the term rigid arrow to describe previous works, specifically, I find a little unclear. In the work by Khorsand they are sensitive to measurements of the change in the length of the magnetisation, which I would argue is not rigid.

→ We thank the reviewer for the thoughtful opinion. As the reviewer mentioned, rigid arrow is insufficient to indicate a change of magnetization under pump laser incidence. So, we changed the term from *rigid arrow model* to *oppositely headed two arrows*.

Reviewer #2 (Remarks to the Author):

I prefer using "Kerr signal" in figures because Kerr angle is a special angle which should be measured by rotating the polarizer till the maximum signal appears, i.e. the Kerr angle is the rotation of polarization of reflected light. Normally we measure the Kerr signal by setting a polarizer at a suitable angle, just detecting the variation of reflected light intensity at a certain angle of polarizer.

→ We thank the reviewer for exact comment. According to the definition of the Kerr angle mentioned by the reviewer, it is appropriate to use the Kerr signal rather than the Kerr angle when presenting the results of our experiments.

Reviewer #3 (Remarks to the Author):

I would like to recommend the revised version for publication in this journal

→ We thank the reviewer for your positive review of our response.